# Unraveling the Complex Interplay between Alpha-Synuclein and Epigenetic Modification

**DOI:** 10.3390/ijms24076645

**Published:** 2023-04-02

**Authors:** Naoto Sugeno, Takafumi Hasegawa

**Affiliations:** Division of Neurology, Department of Neuroscience & Sensory Organs, Tohoku University Graduate School of Medicine, Sendai 980-8574, Japan; sugeno@med.tohoku.ac.jp

**Keywords:** alpha-synuclein, bioinformatics, epigenome, organoids

## Abstract

Alpha-synuclein (αS) is a small, presynaptic neuronal protein encoded by the *SNCA* gene. Point mutations and gene multiplication of *SNCA* cause rare familial forms of Parkinson’s disease (PD). Misfolded αS is cytotoxic and is a component of Lewy bodies, which are a pathological hallmark of PD. Because *SNCA* multiplication is sufficient to cause full-blown PD, gene dosage likely has a strong impact on pathogenesis. In sporadic PD, increased *SNCA* expression resulting from a minor genetic background and various environmental factors may contribute to pathogenesis in a complementary manner. With respect to genetic background, several risk loci neighboring the *SNCA* gene have been identified, and epigenetic alterations, such as CpG methylation and regulatory histone marks, are considered important factors. These alterations synergistically upregulate αS expression and some post-translational modifications of αS facilitate its translocation to the nucleus. Nuclear αS interacts with DNA, histones, and their modifiers to alter epigenetic status; thereby, influencing the stability of neuronal function. Epigenetic changes do not affect the gene itself but can provide an appropriate transcriptional response for neuronal survival through DNA methylation or histone modifications. As a new approach, publicly available RNA sequencing datasets from human midbrain-like organoids may be used to compare transcriptional responses through epigenetic alterations. This informatic approach combined with the vast amount of transcriptomics data will lead to the discovery of novel pathways for the development of disease-modifying therapies for PD.

## 1. Introduction

α-Synuclein (αS) is a relatively small protein with a molecular weight of 14.5 kDa. It consists of 140 amino acids encoded by the *SNCA* gene on chromosome 4q22.1 [1]. The first synuclein cDNA was isolated from the electric organ of *Torpedo californica* and was primarily detected in the nuclear envelope and presynaptic nerve terminals [2]. Thus, the name αS is derived from a combination of the prefixes for synapse (“syn”), nucleus (“nucl”), and the suffix for protein (“ein”). αS is considered an important molecule that triggers the neurodegenerative process in synucleinopathies, including Parkinson’s disease (PD) [3]. Most cases of PD are sporadic; however, less than 10% have a family history and several genes associated with the inherited forms of PD have been identified [4]. *SNCA* was the first familial PD gene (PARK1) discovered, and patients harboring missense mutations exhibit classic adult-onset forms of PD [5]. *SNCA* can cause PD not only through a point mutation, but also by gene multiplication, with the latter designated as *PARK4* [6,7]. These copy number changes correlate with elevated transcript levels of *SNCA* and subsequent increase of αS protein production [6]. In addition, increased *SNCA* mRNA was observed predominantly in the midbrain or substantia nigra of patients with PD [8], while a decrease is detected in other tissues and regions, such as temporal and frontal cortexes and cerebrospinal fluid [9]. It is assumed that this tissue-specific increase in *SNCA* is closely related to neurodegeneration. Thereby, the idea that increased αS expression in the nervous system can cause dopaminergic neuron loss is the rationale for using αS overexpressing cell and animal models for PD studies.

Despite the historical findings of its localization in the nucleus and presynaptic nerve terminals [2], the physiological function of αS has been primarily focused in the synaptic area, where it is involved in regulating synaptic vesicle exocytosis and plasticity [10,11]. Although the majority of αS is localized to the cytoplasm of neurons, expression has been found in other cellular compartments, such as the endoplasmic reticulum [12,13], endosome/lysosomes [14], and mitochondria [15,16]. The transcriptional coactivator PGC-1α is a master regulator of mitochondrial biogenesis and oxidative metabolism [17]. PGC-1α-response genes are significantly associated with PD pathology [18], and interestingly, PGC-1α expression directly influences the oligomerization of αS in cell culture models [19].

Although the pathological processes underlying neurodegenerative diseases are different for every disease, several neurodegenerative molecules are known to function in the nucleus. For example, TAR DNA-binding protein 43 (TDP-43) and fused in sarcoma (FUS), which are associated with amyotrophic lateral sclerosis, are DNA/RNA-binding proteins that contain nuclear localization sequences (NLS) [20,21]. αS does not contain a canonical NLS; however, several studies have indicated that αS is located in the nucleus in the mammalian central nervous system [22]. Moreover, an autopsy revealed that αS-positive glial nuclear and neuronal nuclear inclusions are present in approximately 80% of patients with multiple system atrophy, a type of synucleinopathy [23]. These nuclear inclusions are ubiquitinated, phosphorylated, and composed of fibrillar filaments that are 10–20 nm in diameter [24]. Although it is inconclusive whether this finding is a cause or consequence, the appearance of abnormally modified and structurally altered αS in neural and glial nuclei may be associated with the progression of synucleinopathy, which is a group of neurodegenerative disorders characterized by fibrillary aggregates of αS protein.

The mechanisms by which αS localizes to the nucleus remains elusive. Because of its small size, αS can pass through nuclear pores and does not require transport carriers, such as importin [25]. After internalization into the nucleus, αS may be retained through its interaction with DNA or histones [26,27,28]. Alternatively, several active mechanisms have been suggested to regulate the nuclear translocation of αS, including its interaction with TRIM28 [29] or Ras-related nuclear protein [30]. The function of αS in the nucleus has remained enigmatic, but is becoming clearer with recent progress in epigenetics [31]. In this review, we describe the genetic and epigenetic regulation of *SNCA* transcription and discuss the role of αS in the regulation of epigenetic status and neurodegeneration. In addition, we provide a summary of the currently available high-throughput transcriptome sequencing datasets.

## 2. Interaction between αS and Epigenetic Factors

### 2.1. Transcriptional Regulation of SNCA

Comprehensive genome-wide association studies (GWASs) have examined the association between SNPs and the development of idiopathic PD. Several risk loci in the *SNCA* gene have been identified [32]. A meta-analysis revealed that rs356182 is the most significant SNPs associated with PD [32]; however, it is debatable whether the rs356168 risk variant acts directly on *SNCA* regulation or is involved in a neurodegenerative process unrelated to αS function [33,34]. In addition, the complex polymorphic microsatellite repeat site Rep1, located approximately 10 kb upstream of the *SNCA* transcription start site, has been reported [35]. Longer Rep1 alleles increase *SNCA* gene expression compared with shorter alleles [35]; however, the effect of the Rep allele on disease severity and the risk of cognitive decline remains controversial [36,37,38].

In addition to gene mutations, epigenetic regulation is important to the regulation of *SNCA* expression. DNA methylation occurs on cytosine residues located adjacent to guanine, which are known as CpG sites. The CpG site near the transcription start site, known as the CpG island, and its methylation, represses the transcription of the associated gene. The *SNCA* CpG island is located in intron 1, which is upstream of the initiation codon [39]. It serves as a binding site for several transcription factors that regulate *SNCA* expression [40] (Figure 1). Hypomethylation of this region results in increased expression of *SNCA*, which leads to the accumulation of αS and ultimately, neurotoxicity [41]. In a study of post-mortem brains, a marked reduction in *SNCA* methylation was observed in the substantia nigra, putamen, and cortex of PD patients [42]. Although PD-related alterations in methylation status in intron 1 is controversial [43,44,45,46,47,48,49,50,51], the induction of robust hypermethylation of *SNCA* CpG island results in a reduction in *SNCA* expression, which may be exploited as a therapeutic approach to prevent pathogenic αS accumulation [52]. This intron 1 domain is also associated with histone modification, and another epigenetic modulator, H3K4me3 transcriptional active mark, was more prevalent in the substantia nigra of PD brains [53]. Based on these observations, attempts have been made to regulate epigenetically active histone marks by gene editing in an experimental model of PD. Precisely, a deletion of H3K4me3 by locus-specific editing successfully reduced αS in SH-SY5Y neuronal cells and iPSC-derived dopaminergic neurons [53].

### 2.2. Effect of αS Nuclear Localization

As noted above, αS does not have a canonical NLS; but, post-translational modifications and other factors affect its nuclear localization. In neuronal inclusion bodies (i.e., Lewy bodies) of PD brains, greater than 90% of the αS is highly phosphorylated at serine 129 (S129), which is used as a marker for pathological diagnosis. In vivo experiments using mouse brain and primary cortical neurons have revealed that S129 phosphorylated αS is rapidly translocated to the nucleus of laser-induced focal lesions [54]. Similarly, in H4 human neuroglioma cells, S129 phosphorylated αS exhibited higher affinity toward the nucleus; whereas, downregulated cyclin B1 and E2F transcription factor 8 genes were involved in the cell cycle [55]. Polo-like kinase 2 (PLK2), an enzyme that catalyzes the S129 phosphorylation of αS, may be involved in the nuclear-cytoplasmic shuttling of αS [56]. The presence of S129 phospho-αS may be associated with aging, because S129 phosphorylated αS was observed in the nucleus of aged mice, but not in young A30P human αS transgenic mice [57].

Besides phosphorylation, SUMOylation, which is generated by mature small ubiquitin-related modifiers (SUMO), may occur during the nuclear translocation of αS [58]. Interestingly, distinct αS species translocate from the nucleus to neuronal processes during neuronal differentiation, which suggests that the maturation process of the nervous system may affect subcellular localization [59]. With respect to protein conformational change and aggregation, exposure to human fibrotic αS seed facilitates the formation of intranuclear inclusions in mouse primary cortical neurons [60]. Furthermore, the inoculation of αS preformed fibrils into the stomach wall of wild-type mice resulted in the formation of a small number of nuclear inclusions in the dorsal motor nucleus of the vagal nerve [61]. These results suggest that the progression of PD may contribute to the aberrant accumulation of misfolded αS in neuronal nuclei.

### 2.3. Interplay between Alpha-Synuclein and DNA

#### 2.3.1. Direct Binding to DNA

αS can bind to supercoiled DNA in a conformation-specific manner and alter the bending properties and stability of DNA, which in turn modulates the conformation status of αS itself [62,63,64]. αS also directly binds to a large subset of DNA promoter sequences, including the *PPARGC1A* (PGC-1α) and *NOTCH1* gene promoters, thereby downregulating the transcription of their target genes [55,65,66]. Interestingly, αS colocalizes with DNA damage response components to form discrete foci in the neuronal nuclei and the removal of αS decreases the repair of DNA double-strand breaks [54], suggesting a neuroprotective function of endogenous αS.

#### 2.3.2. Interaction with DNA-Modifying Proteins

DNA methylation, which is an important epigenetic process, is catalyzed by multiple DNA methyltransferases (DNMTs: DNMT1, DNMT2, DNMT3, DNMT3L) through the transfer of a methyl group from *S*-adenyl methionine (SAM) to the fifth carbon of cytosine, resulting in 5mC [67]. During DNA replication, DNMT1 retrieves the DNA methylation prototype from the parent DNA strand and transfers it to the newly synthesized daughter strand [68]. Conversely, DNA demethylation is often catalyzed by enzymes of the ten-eleven translocation (TET) family, which counteract the activity of DNMTs [69]. In recent years, DNA methylation has been a significant area of interest in PD research [70]. Given that CpG methylation profiles associated with PD matched approximately 30% between brain tissue and blood samples [71], specific loci could be identified as candidate biomarkers for PD in peripheral blood mononuclear cells [72]. An early study demonstrated that DNMT1, normally located in the nucleus, was sequestered in the cytoplasm following αS overexpression, causing global DNA hypomethylation and transcriptional activation of downstream genes [73]. A subsequent study also found that altered DNA methylation at CpG sites affected gene expression associated with locomotor behavior and the glutamate signaling pathway [74].

### 2.4. Interplay between Alpha-Synuclein and Histones

A nucleosome is a fundamental unit of chromatin consisting of 147 base pairs of DNA and an octamer of core histone proteins containing two copies of each of the histones: H2A, H2B, H3, and H4 [75]. The amino acid sequences of the histone proteins are conserved across species from *Archaeum* to *Homo sapiens* [76]. The chromosome structure of eukaryotic cells may be divided into two regions: heterochromatin, which is tightly condensed and transcriptionally repressed, and euchromatin, which is untangled and transcriptionally active [76]. The condensation of histones is essentially responsible for the organization of euchromatin and heterochromatin. The chromatin status is defined by the types of post-translational modifications in histone tails, including acetylation, dopaminylation, methylation, phosphorylation, serotonylation, SUMOylation, and ubiquitination [77,78,79]. These histone modifications may alter the affinity to DNA and other histones by altering the surface charge; thereby, reversibly regulating the entry of transcription factors and other transcription-related proteins [76].

#### 2.4.1. Histone Modification via Acetylation

αS binds to the N-terminal flexible tails of histones H3, H4, and H1 [80]. Its fibrillation is accelerated by H1 released from the nucleus during apoptosis [81]. Lysine acetylation is a reversible process, which is post-translational modification that alters the charge of lysine residues and modifies protein structure to influence protein function [82]. Histone acetylation modulates fundamental cell processes, such as transcriptional regulation and chromatin remodeling. The balance between the activities of lysine acetyltransferases (KAT) and histone deacetylases (HDAC) is strictly controlled, but can be disrupted in neurodegenerative diseases, such as PD [83].

KAT catalyzes the transfer of an acetyl group from acetyl-CoA to the *ε*-amino group of an internal lysine residue [84]. Acetylated histones counteract the positive charge on the residue, which prevents DNA-histone interactions and activates transcription, resulting in a loose chromatin state that facilitates transcription. In contrast, histone deacetylation results in a tighter chromatin structure, which suppresses transcriptional activity [84]. Mammalian KATs are classified into two groups according to their cellular localization: Nuclear KATs (type A) and cytoplasmic KATs (type B). Type A KATs are primarily involved in transcriptional regulation and may be further classified into five families: Gcn5-related *N*-acetyltransferase (GNAT), p300/cAMP response element binding (CREB) binding protein (CBP), MYST basal transcription factors, and the nuclear receptor coactivator (NCoA) family [82]. The reverse reaction, deacetylation, is catalyzed by histone deacetylases. To date, 18 mammalian HDACs have been identified and classified into four classes (class I, II, III, and IV) based on their sequence similarity to yeast HDACs. Class I HDACs include HDAC1, 2, 3, and 8; Class II HDACs are subdivided into class IIa (HDAC4, 5, 7, and 9) and IIb (HDAC6 and 10); Class III HDACs are members of the sirtuin family; and class IV HDAC includes only HDAC11 [85]. Of note, nucleosome proteins are not a specific substrate for these KATs and HDACs [86]. For example, class IIa HDACs (HDAC4, HDAC5, and HDAC7) shuttle between the nucleus and the cytoplasm [87]. HDAC4 recognizes a variety of extra-nuclear proteins as substrates, including forkhead transcription factors of the O class (FOXO), myosin heavy chain isoforms (MyHC), PGC-1α, and heat shock cognate 71 kDa (Hsc70) [88].

In general, histone acetylation is associated with gene activation; whereas, the removal of the acetyl mark induces a closed chromatin structure. Several studies suggest that αS reduces histone acetylation, which inhibits the expression of certain genes (Figure 2). Although the intracellular function of αS in neurodegenerative processes remains unclear, nuclear-localized αS increases cytotoxicity [27] whereas cytosolic αS is neuroprotective [89]. Cytosolic αS reduces p300 levels and its KAT activity, resulting in a reduction of histone acetylation in dopaminergic neural cell lines [89]. Alternatively, the A53T mutant αS modulates histone acetylation by interacting with transcriptional adapter 2-α (TADA2a), a component of the major histone acetyltransferase p300/CBP [90].

Class IIa HDACs have NLS and shuttle between the nucleus and cytoplasm [91]. HDAC4, a member of the class IIa HDACs, is abundantly expressed in neurons and accumulates in the nucleus following stimulation with MPTP (1-methyl-4-phenyl-1, 2, 3, 6-tetrahydropyridines), a dopaminergic neurotoxin. Nuclear HDAC4 also mediates cell death in A53T mutant αS-expressing cells by inhibiting CREB and myocyte enhancer factor 2A (MEF2A) [92]. The global HDAC inhibitors, sodium butyrate (NaB) and suberoylanilide hydroxamic acid (SAHA), protect against αS-mediated neurotoxicity in cell and transgenic *Drosophila* models [27]. Decreased H3 acetylation and altered RNA-seq gene expression profiles through αS in LUHMES (Lund human mesencephalic) dopaminergic cells were attenuated by adding NaB, which may be mediated by DNA repair [93]. In contrast, the neuroprotective effects of NaB in PC12 cells are dependent on the activation of PGC-1α via hyperacetylation of its promoter region [94]. Class I HDAC-specific inhibitors and class IIb HDAC6 inhibitors failed to alleviate the αS-induced neurite outgrowth defects in SH-SY5Y cells; whereas, the class IIa HDAC4/5 inhibitor LMK235 successfully promoted neurite outgrowth [95]. The sirtuin family of NAD(+)-dependent class III HDACs are also candidate therapeutic targets in PD [96]. Blockade of sirtuin 2 resulted in a dose-dependent protective effect against αS-induced toxicity [97,98], suggesting the potential therapeutic applications of targeting specific HDACs. The neuroprotective effects of HDAC inhibitors are currently under preclinical investigation as disease-modifying therapy for PD [99]; however, some of their effects may be mediated by mechanisms unrelated to histone acetylation, such as microtubule stabilization [100].

#### 2.4.2. Histone Modification via Methylation

Lysine residues can accept mono-, di-, and tri-methylation (me1, me2, and me3) modifications; whereas, arginine residues accept asymmetric or symmetric di-methylation or mono-methylation. Although histone acetylation usually promotes gene expression, the function of histone methylation depends on the context. Methylation of histone H3 on lysine 4 (H3K4), lysine 36 (H3K36), or lysine 79 (H3K79) is largely responsible for transcriptional activation. In contrast, methylation of histone H3 at lysine 9 (H3K9) and lysine 27 (H3K27) or histone H4 on lysine 20 (H4K20) is often associated with transcriptional repression [101]. Lysine methylation is catalyzed by lysine methyltransferases (KMTs), known as “Writers,”; whereas, histone demethylation is catalyzed by lysine demethylases (KDMs), known as “Erasers”.

Protein arginine methyltransferases (PRMTs) transfer methyl groups from *S*-Adenosylmethionine (SAM) to arginine residues on histone proteins. They are classified into three types based on their catalytic activity [102]. Type I PRMTs (PRMT1, 2, 3, 4, 6, and 8) asymmetrically di-methylate arginine residues (ADMA) whereas type II PRMTs (PRMT5 and PRMT9) symmetrically di-methylate arginine residues (SDMA). Type III (PRMT7) catalyzes only mono-methylated arginine formation (MMA) [102]. The major targets of arginine methylation in histone proteins are histone H3 on arginine 2 (H3R2), arginine 8 (H3R8), and histone H4 on arginine 3 (H4R3). The histone code of arginine methylation is rather complicated. Both ADMA and SDMA are di-methylated, but the asymmetric type, ADMA, is often associated with transcriptional activation, whereas the symmetric type, SDMA, results in transcriptional repression [103]. Therefore, the structural behavior of chromatin differs depending on the type of PRMT acting on the target histone tails. Consequently, SDMA occupying regional histones corresponds to the signature of type II PRMTs.

αS has been found to interact with several epigenetic writers. In an αS yeast model, the altered histone marks including H3K36 di-methylation were distinct from histone marks affected by TDP-43 or FUS [104]. Additionally, overexpression of αS in transgenic *Drosophila* and SH-SY5Y cells resulted in H3K9 di-methylation through upregulation of euchromatic histone lysine *N*-methyltransferase 2 (*EHMT2*) in the presence of retinoic acid [105]. The chromatin immunoprecipitation with antibodies against repressor element-1 (RE1)-silencing transcription factor (REST) inactivated transcription, one of the EHMT2 interacting protein, revealed the repressed downstream genes *SNAP25* and *L1CAM.* SNAP25 is a major component of SNARE complex involved in synaptic function; thereby, these changes may contribute to the synaptic dysfunction that occurs in PD brain [106].

#### 2.4.3. SWI/SNF Chromatin Remodeling Complexes

The SWI/SNF family was first functionally characterized in *Saccharomyces cerevisiae* but is conserved across species throughout eukaryotes. The SWI/SNF family contains an ATPase subunit that utilizes ATP-dependent chromatin remodeling to enhance DNA accessibility during transcription [107]. BAF (Brg1-associated factors) is a mammalian homolog of SWI/SNF, whose central ATPase is composed of SWI/SNF-related, matrix-associated, actin-dependent regulator of chromatin, subfamily A, member 4 (SMARCA4), and SMARCA2. The BAF complexes are not just chromatin remodeling factors, but they can repress or activate gene expression [108]. Interestingly, SMARCA4 harbors SNPs and a meta-analysis yielded *p*-values of 1 × 10^−4^ and 0.05, which may be considered a potential risk for PD, although not in the top 10,000 most significant GWAS results [32,109]. Furthermore, in silico disease-associated gene prediction followed by in vivo *Drosophila* genetic screening identified SMARCA4. Knockdown of Brahma, the *Drosophila* homolog of SMARCA4, in dopaminergic neurons prolonged the lifespan of human *LRRK2* or *SNCA* transgenic *Drosophila* [109]. Another component of the BAF complex, Brg-associated factor 57 (BAF57), was modulated in PC12 cells treated with the dopaminergic neurotoxin, 6-OHDA [110].

## 3. Exploring Transcriptional Modulation Downstream of Epigenetic Regulation: A Disease Model-Based Approach

Recent advances in next-generation sequencing and associated technologies, such as single-cell analysis, have supported innovative evaluation systems including organoid models [111]. These omics techniques are expected to further clarify the pathophysiology of PD involving αS. As a recent trend, many scientific journals require that the omics dataset used for study should be registered in publicly available databases, such as GEO database (https://www.ncbi.nlm.nih.gov/gds), ArrayExpress (https://www.ebi.ac.uk/biostudies/arrayexpress), and DDBJ (https://www.ddbj.nig.ac.jp). Currently, there are limited numbers of reliable “Expression profiling by high-throughput sequencing” datasets associated with *SNCA*; however, if the datasets are enriched by the application of simplified *SNCA* overexpressing cellular and animal models, they may reveal undiscovered aspects of αS, leading to a better understanding of the complex pathogenesis of PD [112,113,114,115].

### 3.1. Human Dopaminergic Cell Lines

SH-SY5Y is a neuroblastoma cell line widely used in PD research [116]. Neurochemically, SH-SY5Y cells are characterized by moderate dopamine-β-hydroxylase activity, basal noradrenaline release, and low tyrosine hydroxylase activity [117,118]. Tyrosine hydroxylase is the rate-limiting enzyme that converts tyrosine to L-dopa, a precursor of dopamine [119]. Dopa is further converted to noradrenaline by dopamine-β-hydroxylase [120]. Thus, the undifferentiated SH-SY5Y cell line may exhibit a catecholaminergic phenotype as it can synthesize both dopamine and noradrenaline [121]. Differentiation induction protocols are required to achieve a mature neuronal phenotype for SH-SY5Y cells. Commonly used protocols include exposure to RA alone or RA treatment followed by incubation with brain-derived neurotrophic factor (BDNF) [116]. A single RA treatment does not repress the expression of REST [105], which is a transcription factor that acts as a repressor for neuronal genes in non-neuronal cells [122]. Sequential treatment with RA and BDNF induces a neuronal morphology with axonal elongation and increased expression of NeuN and tyrosine hydroxylase [123]. Several RNA-seq datasets of SH-SY5Y cells combining RA-induced differentiation and αS overexpression are publicly available [124,125,126] (Table 1). αS binds RA and translocates to the nucleus, where it regulates gene transcription via RA-associated nuclear receptors [124]. Interestingly, RNA-seq revealed that the toxicity of αS in SH-SY5Y cells is enhanced by RA treatment, and the expression levels of PD-related genes such as ATPase cation transporting 13A2 (*ATP13A2*) and PTEN-induced kinase1 (*PINK1*) were also upregulated [124]. Another cell line, the LUHMES cell, is a subclone of human mesencephalic MESC 2.10 cells, which were isolated from the ventral mesencephalon of an eight-week-old fetus and immortalized by introducing the tetracycline-controlled *v-myc* gene [127]. In the Tet system, tetracycline can stop proliferation and growth factors induce differentiation by expressing markers of the neuronal and dopaminergic systems [127]. One RNA-seq dataset of LUHMES cells is available in the GEO database (Table 1). This study detected decreased DNA repair-associated genes include *TOP2A* encodes DNA topoisomerase 2α [93].

### 3.2. Human-Induced Pluripotent Stem Cells

The introduction of human-induced pluripotent stem cell (iPSC) technology, discovered by Takahashi and Yamanaka in 2006, revolutionized modeling of various human diseases [128]. In 2011, midbrain dopaminergic neurons (mDAN) were successfully differentiated from patient-derived iPSCs containing *SNCA* amplification [129,130]. Subsequently, extensive transcriptomic analyses of sporadic PD-derived fibroblasts and associated iPSCs and mDAN revealed the significance of known pathological pathways, such as CREB and PGC1α [131]. These results have increased the reliability of iPSCs-derived neurons as a tool to study the pathogenesis of PD. In addition, RNA-seq analysis of mDAN derived from iPSCs of PD patients carrying a triplication of the *SNCA* locus detected the upregulated dopamine D2-like receptor (D2R) pathway and the G-protein activated inward rectifier potassium channel (GIRK) network [132]. Further transcript analyses showed the decreased *KCNJ6*, a gene encoding GIRK2 that is a marker for dopaminergic neuron in substantia nigra (A9), and these results implicate the epigenetic regulation of receptor-associated genes underlying electrophysiological dysfunction of mDAN [132].

**Table 1 ijms-24-06645-t001:** Publicly available RNA sequencing datasets of cell or animal models expressing *SNCA* or exposure to αS fibril.

Cell Type	Expression	Study Series	References
***Homo sapiens*—High expression of αS**			
SH-SY5Y	pCMV6	GSE186255/GSE183410	[125]
SH-SY5Y wRA	Adeno	GSE149559	[126]
SH-SY5Y wRA	Tet	GSE145804	[124]
LUHMES	lenti	GSE59115	[93]
iPSC-cortical neuron	4copy-I	GSE199349	[133]
iPSC-dopaminergic	4copy-I	GSE163344	[132]
***Homo sapiens*—Treated with αS fibril**			
iPSC-dopaminergic	4copy-g	GSE171999	[114]
***Rattus norvegicus* (Sprague-Dawley)**			
Frontal cortex	BAC-*SNCA*	GSE150646	[112]
***Mus musculus* (C57BL/6N)**			
Striatum	BAC-*SNCA*	GSE116010	[115]
Microglia	mThy1-*SNCA*	GSE158980	[113]

The abbreviations used are as follows: 4copy-I, iPSC line from “Iowa kindred” having 4 copies of *SNCA*; 4copy-g; 4 copies of *SNCA* generated by gene editing using CRISPR-Cas9; αS, alpha-synuclein; Adeno, *SNCA* overexpression by Adeno-X Expression System 2; BAC-*SNCA*, bacterial artificial chromosome transgenic full-length human *SNCA* locus; iPSC, induced pluripotent stem cell; lenti, *SNCA* overexpression by lentiviral vector pWPI; LUHMES, Lund human mesencephalic cell; mThy1, *SNCA* gene driven by the mouse thymus cell antigen 1 promoter; pCMV6, *SNCA* overexpression by pCMV6 vector; Tet, inducible *SNCA* overexpression by pCMV-Tet 3G plasmid; wRA, treatment with retinoic acid.

### 3.3. Organoid Model

Three-dimensional (3D) self-organizing cell structures mimic real organs by exhibiting a well-defined spatial organization that can differentiate into different cell types; thus, enabling physiological studies of cell autonomous and non-cell autonomous pathogenesis of neurodegenerative diseases [134]. Following the establishment of an effective method for differentiating two-dimensional mDAN from pluripotent cells [135], human midbrain-like organoids (hMLOs) were generated [136]. Electrophysiologically active and functionally mature midbrain dopaminergic neurons have also been successfully established [137]. The protocol for hMLOs was repeatedly modified and improved to obtain reproducibility and effectiveness [138,139].

Previously, human embryonic stem cells were used as a source of pluripotent cells; however, in recent years, they have mostly been replaced by iPSCs [140,141,142,143,144]. As an example, a recent study analyzed single-cell RNA sequencing of the human fetal midbrain at 7.5 weeks of gestation and their subsequent differentiation into hMLOs [145]; however, the significance of using fetal midbrain remains uncertain. The initial step in creating hMLOs is midbrain specification using dual SMAD inhibition and Wnt modulation. This is followed by midbrain floor plate induction, in which sonic hedgehog (SHH) and fibroblast growth factor 8 (FGF8) signaling are activated. This allows terminal cell specificity after differentiation, which is mostly restricted to midbrain tissues. Thus, the population of differentiated cells in hMLOs may depend on the quality of the floor plate. Better methods to obtain the equivalent properties of the floor plate cells are needed. For this purpose, human neuroepithelial stem cells (NESC, also known as small molecule neural precursor cells (smNPCs)) [146] were used instead of iPSCs as a starting population. Then, NESC-derived midbrain floor plate neural progenitor cells (mfNPCs) were used for generating hMLOs [147,148]. This protocol revision is expected to improve reproducibility and reduce time. In addition, during the growth phase, cells were embedded into a laminin-based scaffold structure and incubated with conditioned medium containing neurotrophic factors in ultra-low attaching dishes to prevent adhesion to the wall surface using orbital shakers or microfluidics devices [111,149]. Other materials, such as spider-silk or carbon fibers, may be added as basic scaffolds to obtain a larger size or homogeneous cell population [150,151]. When the hMLOs grow beyond a certain size, oxygen and nutrient diffusion to the deeper region is insufficient and a core area composed of dying cells appears [152].

When developing models to mimic PD pathology, hMLOs are exposed to 6-OHDA, MPTP, or trimethylamine *N*-oxide, which is a gut metabolite from the midbrain of elderly and PD patients [134,153,154,155]. Differentiated mDAN from neural stem/precursor cells isolated from hMLOs, rather than crude pluripotent cells, are a stable source for tissue engineering [156]. iPSCs harboring genetic background-derived hMLOs were investigated in iPSCs derived from idiopathic cases of PD [157]. This experimental technique has also been performed in samples with *SNCA* triplication, mutant LRRK2, and other gene mutations associated with familial PD [147,158,159,160,161,162,163,164,165,166,167] (Table 2). hMLOs harboring *SNCA* multiplications achieved intracellular aggregate formation composed by αS, but the neurodegenerative process in this model is currently under investigation [158,160]. Whereas multi-omics analysis of hMLO with *PINK1* mutation suggested dysregulation of autophagic flux, one of the proteolytic systems whose disruption triggers neuronal degeneration in PD [165].

## 4. Conclusions

Although several genetic abnormalities, including point mutations or gene multiplications of *SNCA*, can cause familial PD as a monogenic disorder, the majority of PD cases encountered in clinical practice are sporadic. This suggests that multiple hits of minor genetic and environmental factors can trigger the pathogenesis of PD [168,169]. With respect to the genetic background, several risk loci neighboring *SNCA* have been identified and epigenetic alterations, such as CpG methylation and H3K4me3, are also important factors. These genetic and epigenetic alterations can upregulate αS expression, and some post-translational modifications promote the nuclear translocation of αS. Nuclear αS interacts with DNA, histones, and their modifiers to alter epigenetic status and disrupt the homeostasis of neural function. To counteract these events, advances in gene editing techniques may provide direct access to the central nervous system and low-risk alleles that prevent neurodegeneration. However, the art of molecular biology is not yet mature enough to promote the budding of gene editing, and ethical issues remain an obstacle. On the other hand, epigenetic approaches have the advantage that they do not affect the gene itself; whereas, DNA methylation and histone modifications are reversible. The hMLO model is expected to be a powerful model for disease research and drug discovery because it spatially recapitulates neuron-to-glia interactions and the genetic background in *Homo sapiens*. Recently, a series of publicly available RNA-seq datasets of hMLOs were published, making it easier to compare the transcriptional response through epigenetic alterations. Epigenetic modification is an attractive field in terms of drug repositioning because many drugs targeting HDACs and DNMTs have been developed and some have been commercialized for cancer therapy. This review will be helpful to young researchers who are interested in the function of nuclear αS and epigenetic modification as well as to those who wish to expand their research in this area.

## Figures and Tables

**Figure 1 ijms-24-06645-f001:**
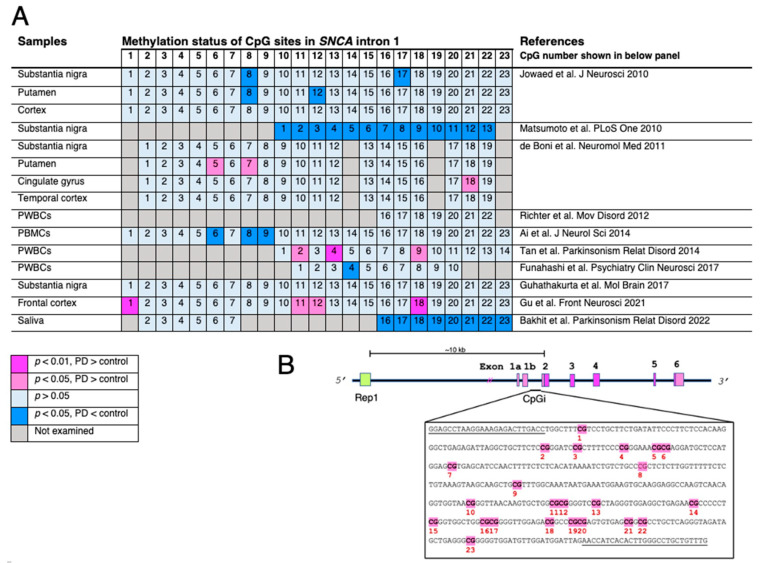
Changes in methylation status of CpG islands located in *SNCA* intron 1. (**A**) The indexed number of CpG sites is based on the report by Jowaed et al. [42,43,44,45,46,47,48,49,50,51]. The numbering of CpG sites is not identical in each study and is, therefore, corrected and displayed accordingly. Differences in methylation rates of CpG sites are compared between Parkinson’s disease (PD) or dementia with Lewy bodies (DLB) and healthy subjects or disease controls. The analyses are varied, but do not show consistent results. Nevertheless, CpG numbers 8 and 18 are likely more sensitive to the methylation modulating system or nuclear environment. (**B**) Schematic presentation of the human *SNCA* gene. Rep1, a dinucleotide repeat site, the length of which affects *SNCA* expression, located approximately 10 kb upstream from the start codon in exon 2. The CpG island located in intron 1 includes 23 CpG sites. Primer sequences for pyrosequencing by Jowaed et al. are underlined. The abbreviations used are as follows: PWBCs, peripheral white blood cells; PBMC, peripheral blood mononuclear cells.

**Figure 2 ijms-24-06645-f002:**
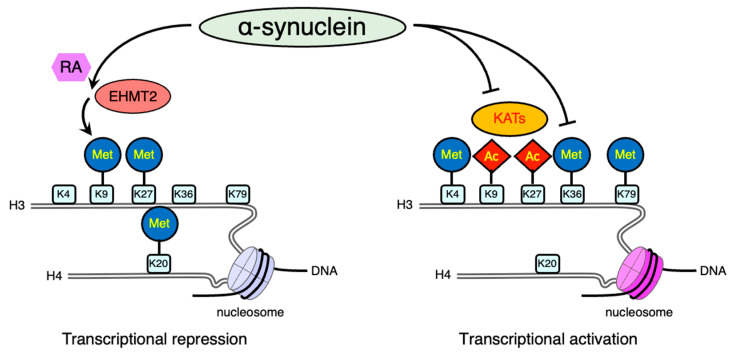
Schematic illustration of post-translational modifications of histone proteins affected by alpha-synuclein (αS). Lysine methylation has different functions depending on its residues. Methylation (Met) of H3 lysine 9 (K9), lysine 27 (K27), and H4 lysine 20 (K20) is often associated with transcriptional repression. In contrast, the methylation of H3 lysine 4 (K4), lysine 36 (K36), or lysine 79 (K79) is largely responsible for transcriptional activation. In addition, histone acetylation (Ac) usually promotes gene expression. Combined stimulation with αS and retinoic acid (RA) enhances K9 methylation of H3 through the activation of euchromatic histone lysine N-methyltransferase 2 (EHMT2). In addition, the inactivation of lysine acetyltransferases (KATs) decreases histone acetylation. The other transcriptional active mark, K36 methylation of H3, is also downregulated by αS. Both increased repressive signals and disruption of active marks result in the transcriptional repression of downstream genes.

**Table 2 ijms-24-06645-t002:** List of reported human midbrain-like organoids.

Starting Material	Additional Procedure	Study Series	Study Type	References
**Development of human Midbrain-like organoids (hMLOs)**				
hESC (HS415)		n.a.	n.a.	[136]
hESC (H9)		E-MTAB-4868	bulk RNA-seq	[137]
hESC (H9)		n.a.	n.a.	[142]
hESC	Brainstem organoids	DRA009864	bulk RNA-seq	[140]
		GSE145306	scRNA-seq	
hESC (H9)	hMLO derived NSCs	GSE142505	bulk RNA-seq	[156]
Fetal midbrain		GSE192405	scRNA-seq	[145]
iPSC		n.a. in hMLOs	n.a.	[144]
smNPC		GSE119060	bulk RNA-seq	[143]
NESC		n.a.	n.a.	[141]
NESC		n.a.	n.a.	[152]
mfNPC		LCSB	scRNA-seq	[148]
**With minor modification**				
hESC	spider-silk	GSE168323	scRNA-seq	[150]
hESC	TH-TdTomato	n.a.	n.a.	[138]
iPSC	carbon fiber	n.a.	n.a.	[151]
iPSC	sevoflurane	n.a.	n.a.	[139]
NESC	millifluidic culture	n.a.	n.a.	[149]
**Parkinson’s disease model**				
iPSC	PD patient derived	n.a.	n.a.	[157]
mfNPC	6-OHDA	n.a.	n.a.	[134]
NESC	6-OHDA	n.a.	n.a.	[153]
hESC	MPTP	n.a.	n.a.	[154]
hESC	TMAO	n.a.	n.a.	[155]
iPSC	*SNCA* triplication	GSE186780	scRNA-seq	[158]
mfNPC	*SNCA* triplication	LCSB	bulk RNA-seq	[159]
hESC	*GBA^−^*^/−^, *SNCA*	E-MTAB-7302	bulk RNA-seq	[160]
hESC	*LRRK2*-G2019S	GSE125234	Array	[161]
mfNPC	*LRRK2*-G2019S	GSE127967	Array	[147]
NESC	*LRRK2*-G2019S	GSE128040	scRNA-seq	[162]
mfNPC	*LRRK2*-G2019S	GSE133894	scRNA-seq	[163]
NESC	*PINK1^−^*^/−^ (TALEN)	n.a.	n.a.	[164]
NESC	*PINK1* Q456X, I368N	n.a.	n.a.	[165]
NESC	*PRKN* mutations	n.a.	n.a.	[166]
hESC	*PRKN^−^*^/−^, *PARK7^−^*^/−^, and *ATP13A2^−^*^/−^	GSE140076	bulk RNA-seq	[167]

The abbreviations used are as follows: 6-OHDA, 6-hydroxydopamine; Array, Expression profiling by array; *ATP13A2*, ATPase cation transporting 13A2 gene responsible for *PARK9*; DRA*, DDBJ (https://www.ddbj.nig.ac.jp); E-MTB-*, ArrayExpress (https://www.ebi.ac.uk/biostudies/arrayexpress); GSE*, GEO database (https://www.ncbi.nlm.nih.gov); hESC, human embryonic stem cell; iPSC, induced pluripotent stem cell; LCSB, institute domain of Luxembourg Center for Systems Biomedicine (https://r3lab.uni.lu/); *LRRK2*-G2019S, heterogeneous G2019S mutation of Leucine-rich repeat kinase 2 (*LRRK2*); mfNPC, midbrain floor plate neural progenitor cell; MPTP, 1-methyl-4-phenyl-1, 2, 3, 6-tetrahydropyridine; n.a., not applicable; NESC, neuroepithelial stem cell (positive for SOX1, SOX2, PAX6, and nestin); *PRKN*, *Parkin* gene responsible for *PARK2*; scRNA-seq, single-cell RNA sequencing; smNPC, small molecule neural precursor cell, the same as NESC; TALEN, Transcription activator-like effector nucleases; TMAO: trimethylamine *N*-oxide; TH-TdTomato, targeting vector to tyrosine hydroxylase gene locus for induction of P2A-fused tdTomato.

## Data Availability

Data sharing is not applicable to this article.

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
