# Peer review of "Unraveling the Complex Interplay between Alpha-Synuclein and Epigenetic Modification"

_ijms, 2023, doi:10.3390/ijms24076645_

Round 1

Reviewer 1 Report

Review of a manuscript “Unraveling the complex interplay between alpha-synuclein and epigenetic modification” by Naoto Sugeno and coauthors.

α-Synuclein is a main player in Parkinson’s disease and several other disorders collectively called “synucleinopathies”. In addition to its role in the formation of toxic aggregates and inclusions, α-synuclein plays a regulatory function in epigenetic mechanisms and other signaling pathways.  The role of post-translational modifications of α-synuclein facilitating its translocation to the nucleus and effect in gene expression regulation require further investigations. The authors describe the genetic and epigenetic regulation of α-synuclein transcription and discuss the role of α-synuclein in the regulation of epigenetic status and neurodegeneration. This is an important area of biomedical science and the results presented in the review will be interesting for the readership of the journal.

The following corrections and additions should be made.

Introduction.

Line 35. “Most cases of PD are sporadic; however, less than 10% have a family history and several genes associated with the inherited forms of PD have been identified.” After this sentence a reference(s) should be added.

Line 38. “SNCA can cause PD not only through a point mutation, but also by gene multiplication…”

The authors should add other cases of α-synuclein multiplication, for example, Farrer et al., (2004) Comparison of kindreds with parkinsonism and alpha-synuclein genomic multiplications. Ann Neurol 55: 174–179

 Line 68. “After internalization into the nucleus, αS may be retained through its interaction with DNA or histones [22, 23].” The authors should add a related reference: “Protein-DNA interaction: One step closer to understanding the mechanism of neurodegeneration.” J Neurosci Res. 2019 Apr;97(4):391-392. doi: 10.1002/jnr.24346.

Line 175. “The chromosome structure of eukaryotic cells may be divided into two regions: heterochromatin, which is tightly condensed and transcriptionally repressed, and euchromatin, which is untangled and transcriptionally active.”  Reference should be given after this sentence.

Line 281-282: ”In an αS yeast model, observed altered histone marks include H3K36 di-methylation were distinct from histone marks affected by TDP-43 or FUS [99].” The sense of this sentence is unclear. May be the authors want to say: ”In an αS yeast model, altered histone marks including H3K36 di-methylation were distinct from histone marks affected by TDP-43 or FUS [99].”?

Lines 322-323 :”This cell line is a subclone of SK-N-SH, which was established in 1970 by culturing a metastatic neuroblastoma from a bone marrow biopsy of a 4-year-old girl [112].” These details are irrelevant to the main topic of the manuscript and may be deleted.  

3.3. Organoid model. The content of this section is not directly to the main focus of the manuscript. It can be converted to a more abbreviated form.  

 Conclusion. “Although several genetic abnormalities, including SNCA”… the authors should be more specific about “abnormalities”. Do they mean “mutations, gene duplication, etc." ?

Author Response

Thank you very much for your letter of Mar 17, 2023, concerning our manuscript entitled in “Unraveling the complex interplay between alpha-synuclein and epigenetic modification” (manuscript ID ijms-2287678) together with the comments from the reviewers. We appreciate the reviewers’ constructive comments and are now resubmitting the revised manuscript. Upon reading the reviewers’ comments carefully, we have modified the sentences.

We believe that the revision process has significantly improved the quality of the paper. The point-by-point responses to the reviewer’s comments are listed as follows:

Reviewer 1

Introduction.

Line 35. “Most cases of PD are sporadic; however, less than 10% have a family history and several genes associated with the inherited forms of PD have been identified.” After this sentence a reference(s) should be added.

Response: We agree with the reviewer's comment. We added a reference in Line 36.

Line 38. “SNCA can cause PD not only through a point mutation, but also by gene multiplication…”

The authors should add other cases of α-synuclein multiplication, for example, Farrer et al., (2004) Comparison of kindreds with parkinsonism and alpha-synuclein genomic multiplications. Ann Neurol 55: 174–179

Response: We agree with the reviewer's comment. We added the reference in Line 40.

 Line 68. “After internalization into the nucleus, αS may be retained through its interaction with DNA or histones [22, 23].” The authors should add a related reference: “Protein-DNA interaction: One step closer to understanding the mechanism of neurodegeneration.” J Neurosci Res. 2019 Apr;97(4):391-392. doi: 10.1002/jnr.24346.

Response: We agree with the reviewer's comment. We added the reference in Line 76.

Line 175. “The chromosome structure of eukaryotic cells may be divided into two regions: heterochromatin, which is tightly condensed and transcriptionally repressed, and euchromatin, which is untangled and transcriptionally active.”  Reference should be given after this sentence.

Response: We agree with the reviewer's comment. We added the reference in Line 188.

Line 281-282: ”In an αS yeast model, observed altered histone marks include H3K36 di-methylation were distinct from histone marks affected by TDP-43 or FUS [99].” The sense of this sentence is unclear. May be the authors want to say: ”In an αS yeast model, altered histone marks including H3K36 di-methylation were distinct from histone marks affected by TDP-43 or FUS [99].”?

Response: We totally agree with the reviewer's comment. We corrected the sentence indicated in Line 291-293.

Lines 322-323 :”This cell line is a subclone of SK-N-SH, which was established in 1970 by culturing a metastatic neuroblastoma from a bone marrow biopsy of a 4-year-old girl [112].” These details are irrelevant to the main topic of the manuscript and may be deleted.  

Response: We agree with the reviewer's comment. We deleted the sentence.

3.3. Organoid model. The content of this section is not directly to the main focus of the manuscript. It can be converted to a more abbreviated form.  

Response: To make a connection with the previous part of the manuscript, we have added results on some transcription adjustments obtained from the hMLOs model of Parkinson’s disease in Line 429-434. For the benefit of readers using the datasets in the list, we have emphasized that each hMLO has different characteristics depending on how it is manufactured. For this reason, the history of hMLO is described somewhat too accurately.

 Conclusion. “Although several genetic abnormalities, including SNCA”… the authors should be more specific about “abnormalities”. Do they mean “mutations, gene duplication, etc." ?

Response: We added the specific gene abnormalities in that sentence in Line 446-447.

Reviewer 2 Report

The Review article by Sugeno, Nakamura & Hasegawa titled “Unraveling the complex interplay between alpha-synuclein and epigenetic modification” is a nice, comprehensive and concise overview of the aforementioned subject. The manuscript is very-well organized, elegantly written and easy to follow. Language is concise and on appropriate level, with only minor errors/typos. The review on the current literature is extensive and suitably put into context. The included graphical work is helpful in transferring key concepts of the review.

I have no major considerations against the content or the structure of the manuscript. Here are some minor points which may help the authors to improve the review even further:

11)      The authors make a clear a well-grounded point that α-Syn should be considered as a gene regulatory factor, since it’s association with the nucleus, DNA and the transcriptional regulation machinery are more than well-described. However, the authors do not elaborate on what gene expression changes are caused by α-Syn, even if they list such studies in a table. It would be nice if the authors summarized (shortly) for the reader what is known so far from these studies.

22) The same is valid for the last part concerning the iPSC and hMLo models: the authors give a very nice background crash course of the models and list SNCA-related studies with such models, but do not elaborate on the major conclusions from these studies. A couple of sentences will be appropriate I think to summarize what knowledge about SNCA and epigenetics/transcriptional regulation was gained from these models so far. Otherwise the part of the review dealing with these models is only loosely tied to the first part of the MS.   

33) Also along the same lines, the rather long part concerning histone modification gives some evidence that α-Syn interacts with histone modifications, but provide very little information on what the consequences were (if specified in the studies).

44) The part concerning SNCA transcriptional regulation discusses DNA methylation, but does not mention histone marks and chromatin accessibility studies at the SNCA gene, which are complimentary to the DNA methylation. Even if such studies are not performed yet, the authors should I think at least mention this.

55) Related to the last point, since the first part of the manuscript elaborates on the epigenetic/transcriptional regulation of SNCA, it would be best if the authors start this study by specifying more concisely what is know of increased SNCA expression in sporadic PD patients in different tissues, and if such gene expression studies have been put into epigenetic context so far by a multiomic approach, e.g. by simultaneous measurement of chromatin accessibility and gene expression.

Minor grammatical/stylistic errors and typos:

-      P2L76: What kind of studies? Transcriptome? WGS? etc, high-throughput sequencing is a very broad term

-          P2L96 “…in a study of postmortem brains (not ‘a postmortem brains’)

-          P4L150: “Interestingly, aSyn colocalizes”,  not “aSyn has colocalizes”

-          P6L188: “Lysine acetylation is a reversible PROCESS, which…”

-          P6L230: “Enhanced stimulation with aSyn and retinoic acid…”, not “of aSyn and retinoic acid”

-          P7:268: PRMT is not abbreviated on first use

-          P7L275: “The histone CODE”,  not “cord”

-          P7L282: “THE observed altered histone marks,  includING H3K36 di-methylation, were distinct from histone..”

-          P7L284; “resulted IN..”

-          P7L288: “revealed”, not “reveled”

-          P11L440 “to be a powerful model for disease research”, not “in for disease research”

Author Response

Thank you very much for your letter of Mar 17, 2023, concerning our manuscript entitled in “Unraveling the complex interplay between alpha-synuclein and epigenetic modification” (manuscript ID ijms-2287678) together with the comments from the reviewers. We appreciate the reviewers’ constructive comments and are now resubmitting the revised manuscript. Upon reading the reviewers’ comments carefully, we have modified the sentences.

We believe that the revision process has significantly improved the quality of the paper. The point-by-point responses to the reviewer’s comments are listed as follows:

11)      The authors make a clear a well-grounded point that α-Syn should be considered as a gene regulatory factor, since it’s association with the nucleus, DNA and the transcriptional regulation machinery are more than well-described. However, the authors do not elaborate on what gene expression changes are caused by α-Syn, even if they list such studies in a table. It would be nice if the authors summarized (shortly) for the reader what is known so far from these studies. 

Response: We added the lines about downstream genes of SH-SY5Y (Line 350-354), and of LUHMES (L360-361).

22) The same is valid for the last part concerning the iPSC and hMLo models: the authors give a very nice background crash course of the models and list SNCA-related studies with such models, but do not elaborate on the major conclusions from these studies. A couple of sentences will be appropriate I think to summarize what knowledge about SNCA and epigenetics/transcriptional regulation was gained from these models so far.Otherwise the part of the review dealing with these models is only loosely tied to the first part of the MS.   

Response: We added the sentences summarizing the transcriptional alterations observed in iPSC and hMLOs models in Line 372-376, and 429-434.

33) Also along the same lines, the rather long part concerning histone modification gives some evidence that α-Syn interacts with histone modifications, but provide very little information on what the consequences were (if specified in the studies). 

Response: We discussed about the histone acetylation via αS and HDAC inhibitors can counteract αS-mediated histone modification in L251-268. αS mediated-histone methylation was described in Line 294-304.

44) The part concerning SNCA transcriptional regulation discusses DNA methylation, but does not mention histone marks and chromatin accessibility studies at the SNCA gene, which are complimentary to the DNA methylation. Even if such studies are not performed yet, the authors should I think at least mention this. 

Response: We appreciate the reviewer’s suggestion because transcriptional regulation of SNCA by histone modification was not mentioned in initial manuscript. We added text indicating that SNCA intron 1 is occupied by the H3K4me3 histone mark in Line 108-111.

55) Related to the last point, since the first part of the manuscript elaborates on the epigenetic/transcriptional regulation of SNCA, it would be best if the authors start this study by specifying more concisely what is know of increased SNCA expression in sporadic PD patients in different tissues, and if such gene expression studies have been put into epigenetic context so far by a multiomic approach, e.g. by simultaneous measurement of chromatin accessibility and gene expression. 

Response: We described SNCA is abundantly expressed in neuron in L52. We also mentioned about SNCA expression in sporadic PD patients in different tissues in Line 41-46.

Minor grammatical/stylistic errors and typos: 

-      P2L76: What kind of studies? Transcriptome? WGS? etc, high-throughput sequencing is a very broad term 

Response: They were transcriptome study. We added this to the sensence in Line 82.

-          P2L96 “…in a study of postmortem brains (not ‘a postmortem brains’) 

Response: We corrected the sentence in Line 103-104.

-          P4L150: “Interestingly, aSyn colocalizes”,  not “aSyn has colocalizes” 

Response: We corrected the sentence in Line 160.

-          P6L188: “Lysine acetylation is a reversible PROCESS, which…” 

Response: We corrected the sentence in Line 198.

-          P6L230: “Enhanced stimulation with aSyn and retinoic acid…”, not “of aSyn and retinoic acid” 

Response: We corrected the sentence in Line 240.

-          P7:268: PRMT is not abbreviated on first use 

Response: We added the formal description of PRMT on first use in Line 278.

-          P7L275: “The histone CODE”,  not “cord” 

Response: We appreciated the review’s kind comment. We corrected the word in Line 285.

-          P7L282: “THE observed altered histone marks,  includING H3K36 di-methylation, were distinct from histone..” 

Response: We corrected the sentence in Line 292-294.

-          P7L284; “resulted IN..” 

Response: We corrected the sentence in Line 295.

-          P7L288: “revealed”, not “reveled” 

Response: We corrected the word in Line 299.

-          P11L440 “to be a powerful model for disease research”, not “in for disease research”

Response: We appreciated the comment. We corrected the sentence in Line 463.